# KNOWLEDGE FLOW: IMPROVE UPON YOUR TEACHERS

**Iou-Jen Liu, Jian Peng, Alexander G. Schwing**
University of Illinois at Urbana-Champaign
`{iliu3, jpeng, aschwing}@illinois.edu`

## ABSTRACT

A zoo of deep nets is available these days for almost any given task, and it is increasingly unclear which net to start with when addressing a new task, or which net to use as an initialization for fine-tuning a new model. To address this issue, in this paper, we develop *knowledge flow* which moves 'knowledge' from *multiple* deep nets, referred to as teachers, to a new deep net model, called the student. The structure of the teachers and the student can differ arbitrarily and they can be trained on entirely different tasks with different output spaces too. Upon training with *knowledge flow* the student is independent of the teachers. We demonstrate our approach on a variety of supervised and reinforcement learning tasks, outperforming fine-tuning and other 'knowledge exchange' methods.

## 1 INTRODUCTION

Research communities have amassed a sizable number of deep net architectures for different tasks, and new ones are added almost daily. Some of those architectures are trained from scratch while others are fine-tuned, *i.e.*, before training, their weights are initialized using a structurally similar deep net which was trained on different data.

Beyond fine-tuning, particularly in reinforcement learning, teachers have also been considered in one way or another by Rusu et al. (2016b); Fernando et al. (2017); Wang et al. (2017); Li & Hoiem (2016); Bengio et al. (2009); Patel et al. (2015); Chen & Liu (2016); Teh et al. (2017); Parisotto et al. (2016). For instance, progressive neural net (Rusu et al., 2016b) keeps multiple teachers during both training and inference, and learns to extract useful features from the teachers for a new target task. PathNet (Fernando et al., 2017) uses genetic algorithms to choose pathways from a giant network for learning new tasks. 'Growing a Brain' (Wang et al., 2017) fine-tunes a neural network while growing the network's capacity (wider or deeper layers). Actor-mimic (Parisotto et al., 2016) pre-trains a big model on multiple source tasks, then the big model is used as a weight initialization for a new model which will be trained on a new target task. Knowledge distillation (Hinton et al., 2015) distills knowledge from a large ensemble of models to a smaller student model.

However, all the aforementioned techniques have limitations. For example, progressive neural net models (Rusu et al., 2016b) grow with the number of teachers. This large number of parameters limits the number of teachers a progressive neural net can handle, and largely increases the training and testing time. In PathNet (Fernando et al., 2017), searching over a big network for pathways is computationally intensive. For fine-tuning based methods such as 'Growing a Brain' (Wang et al., 2017) and actor-mimic (Parisotto et al., 2016), only one pretrained model can be used at a time. Hence, their performance heavily relies on the chosen pretrained model.

To address these shortcomings, we develop *knowledge flow* which moves 'knowledge' of multiple teachers when training a student. Irrespective of how many teachers we use, the student is guaranteed to become independent at the final stage of training and the size of the resulting student net remains constant. In addition, our framework makes no restrictions on the deep net size of the teacher and student, which provides flexibility in choosing teacher models. Importantly, our approach is applicable to a variety of tasks from reinforcement learning to fully-supervised training.

We evaluate *knowledge flow* on a variety of tasks from reinforcement learning to fully-supervised learning. In particular, we follow Rusu et al. (2016b); Fernando et al. (2017) and compare on the same

Atari games. In addition, we also observed significant top-1 error rate improvements on supervised learning datasets, *i.e.*, CIFAR-10, and CIFAR-100.

## 2 BACKGROUND

*Knowledge flow* is applicable to a variety of settings from supervised learning to reinforcement learning, which we briefly review to introduce notation.

**Supervised Learning** recovers the parameters $\theta$ of a mapping $f_\theta : \mathcal{X} \to \mathcal{Y}$ from data space $\mathcal{X}$ to output space $\mathcal{Y}$. To this end, a dataset $D = \{(x_i, y_i)\}_{i=1}^n$ containing $n$ pairs $(x_i, y_i)$ (assumed to be sampled i.i.d.) is used, where $x_i \in \mathcal{X}$ and $y_i \in \mathcal{Y}$. Given this dataset, the parameters $\theta$ of the mapping $f_\theta$ are learned by minimizing a loss function $\ell_{(x,y)}(\theta)$ composed of a regularization term $R(\theta)$ and an empirical risk $\ell(y, f_\theta(x))$ which compares groundtruth label $y$ and prediction $f_\theta(x)$. The parameters $\theta$ are obtained by optimizing the following program:

$$\min_\theta \mathbb{E}_{(x,y) \sim D}[\ell_{(x,y)}(\theta)] := \mathbb{E}_{(x,y) \sim D}[\ell(y, f_\theta(x))] + R(\theta). \tag{1}$$

Hereby, the mapping $f_\theta$ is obtained by maximizing the logits or a corresponding probability distribution $\hat{f}_\theta(y|x)$, *i.e.*, $f_\theta = \arg\max_{y \in \mathcal{Y}} \hat{f}_\theta(y|x)$. Here and below let the hat ('$\hat{\cdot}$') indicate probability distributions over appropriate domains.

**Reinforcement Learning** considers an agent interacting with an environment according to a policy $\pi_{\theta_\pi} : \mathcal{X} \to \mathcal{A}$ which maps a state $x_t \in \mathcal{X}$ to an action $a_t \in \mathcal{A}$ at time $t$. The policy depends on the parameters $\theta_\pi$. After performing action $a_t$, the agent observes the next state $x_{t+1}$ and receives a scalar reward $r_t$. The discounted return at time $t$ is defined as $R_t = \sum_{k=0}^\infty \gamma^k r_{t+k}$, where $\gamma$ is the discount factor. The expected future reward when observing state $x$ and when following policy $\pi_{\theta_\pi}$ is defined as $V^{\pi_{\theta_\pi}}(x_t) = \mathbb{E}_{\tau \sim \pi_{\theta_\pi}}[R_t|x_t]$, where $\tau = \{(x_t, a_t, r_t), (x_{t+1}, a_{t+1}, r_{t+1}), \ldots\}$ is a trajectory generated by following $\pi_{\theta_\pi}$ from state $x_t$.

The goal of reinforcement learning is to find a policy that maximizes the expected future reward from each state $x_t$. Without loss of generality, in this paper, we follow the asynchronous advantage actor-critic (A3C) formulation (Mnih et al., 2016). In A3C, the policy mapping $\pi_{\theta_\pi}(x) = \arg\max_{a \in \mathcal{A}} \hat{\pi}_{\theta_\pi}(a|x)$ is obtained from a probability distribution over states, where $\hat{\pi}_{\theta_\pi}(a|x)$ is modeled by a deep net with parameters $\theta_\pi$. The value function is also approximated by a deep net $V_{\theta_v}(x)$, having parameters $\theta_v$.

To optimize the policy parameters $\theta_\pi$ given a state $x_t$, a loss function based on a scaled negative log-likelihood and a negative entropy regularizer is common:

$$\ell_\pi^\tau(\theta_\pi) = \frac{1}{|\tau|} \sum_{t \in \tau} \left[ -\log \hat{\pi}_{\theta_\pi}(a_t|x_t)(R_t - V_{\theta_v}(x_t)) - \beta H(\hat{\pi}_{\theta_\pi}(\cdot|x_t)) \right].$$

Hereby, $R_t = \sum_{i=0}^{k-1} \gamma^i r_{t+i} + \gamma^k V_{\theta_v}(x_{t+k})$ is the empirical $k$-step return obtained when starting in state $x_t$, and $|\tau|$ is the length of the trajectory $\tau$ generated by following $\pi_{\theta_\pi}$. The scalar $\beta \geq 0$ is a user-specified constant, and $H(\hat{\pi}_{\theta_\pi}(\cdot|x_t))$ is the entropy function, which encourages exploration by favoring a uniform probability distribution $\hat{\pi}_{\theta_\pi}(a|x)$. To optimize the value function $V_{\theta_v}$, it is common to use the squared loss $\ell_v^\tau(\theta_v) = \frac{1}{2|\tau|} \sum_{t \in \tau} (R_t - V_{\theta_v}(x_t))^2$.

By minimizing the empirical expectation of $\ell_\pi^\tau(\theta_\pi)$ and $\ell_v^\tau(\theta_v)$, *i.e.*, by addressing

$$\min_{\theta_\pi} \mathbb{E}_{\tau \sim \pi_{\theta_\pi}}[\ell_\pi^\tau(\theta_\pi)], \quad \text{and} \quad \min_{\theta_v} \mathbb{E}_{\tau \sim \pi_{\theta_\pi}}[\ell_v^\tau(\theta_v)], \tag{2}$$

alternatingly, we learn a policy and a value function that maximize expected return.

## 3 KNOWLEDGE FLOW

Instead of optimizing the programs given in Eq. (1) and Eq. (2) from scratch, the aforementioned warm-start techniques (see Sec. 5 for more) are applicable. To address their mentioned shortcomings, we propose *knowledge flow*, a framework that moves 'knowledge' from an arbitrary number of deep nets, henceforth referred to as 'teachers' to a deep net under training, called the 'student.'

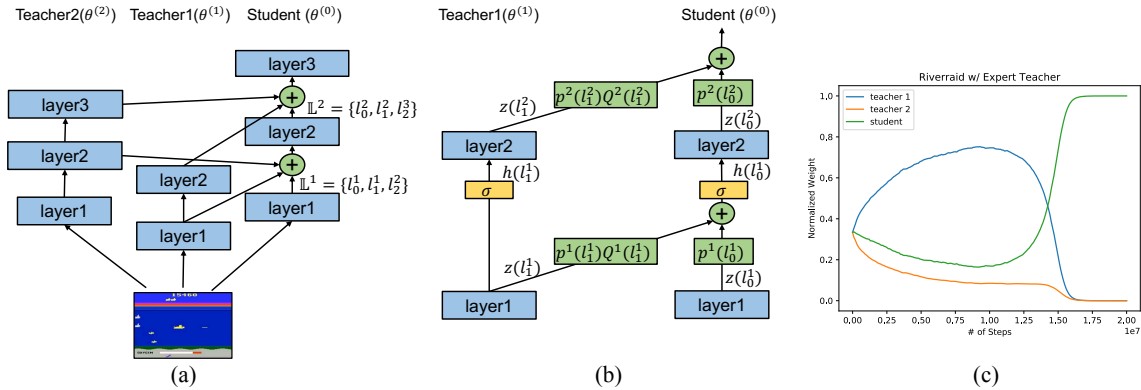

Figure 1: (a) Example of a two-teacher *knowledge flow*. (b) Deep net transformation of *knowledge flow*. (c) Average normalized weights for teachers' and the student's layers. At the beginning of training, the student heavily relies on teacher one. As training progresses, teacher one's weight decreases, and the student's weight increases until the student is eventually independent.

## 3.1 OVERVIEW

*Knowledge flow* is outlined on example deep nets in Fig. 1 (a,b). We train the parameters of the student net which are randomly initialized. To this end we take advantage of teachers, whose parameters are fixed and obtained from pre-trained models on different source tasks by different algorithms. For example, for reinforcement learning, we may consider teachers trained by A3C (Mnih et al., 2016), A2C (Dhariwal et al., 2017) or DQN (Mnih et al., 2015).

'Knowledge' of multiple teachers is transferred to a student by adding transformed and scaled intermediate representations from the teacher deep nets to the student net. To achieve this, we modify the student net, *i.e.*, $f_\theta$ in the supervised setting and $\pi_{\theta_\pi}(a|x)$, $V_{\theta_v}(x)$ in the reinforcement learning case. We add teacher representations which are transformed by multiplication with a trainable matrix Q and scaled via a weight $p_w$ that is normalized to sum to one for each student layer and parameterized via trainable parameters $w$. The normalized weights encode which of the teachers' or the student's representation to trust at every layer of the student net. Note that a teacher can help the student at different levels of abstraction with input from different levels of its net.

Importantly, after training, the student model should perform well on the target task without relying on teachers. To achieve this, as training progresses, we increasingly encourage a high normalized weight on the student representation, which forces the student to eventually capture all the 'knowledge.' Due to the trainable scaling, at an early stage of training, we observe the student to rely heavily on the 'knowledge' of the teacher to quickly obtain better performance. However, as training proceeds, the student is encouraged to become more and more independent. During final stages of training, the student will no longer be able to rely on teachers, which ensures that the student has learned to master the desired task on its own. This is observed in Fig. 1 (c).

To formally encourage this successive transfer we introduce two additional loss functions. The first, referred to as the dependency loss $\ell_{\text{dep}}(w)$, captures how much a student relies on teachers. It depends on the weight vector $w$ which encodes the strength of the coupling. The second one ensures that a student's behavior doesn't change rapidly when the teachers' influence decreases. We use loss $\ell_{\text{KL}}(\cdot,\cdot)$ to capture the change.

By combining student net modifications and additional loss terms, for the supervised task we obtain

$$\min_{\theta,w,Q} \mathbb{E}_{(x,y)}[\tilde{\ell}_{(x,y)}(\theta,w,Q) + \lambda_1\ell_{\text{dep}}(w) + \lambda_2\ell_{\text{KL}}(\tilde{\tilde{f}}_\theta,\tilde{\tilde{f}}_{\theta_{\text{old}}})], \tag{3}$$

and for reinforcement learning the transformed program reads as follows:

$$\begin{cases} \min_{\theta_\pi,w,Q} \mathbb{E}_{\tau\sim\tilde{\pi}_{\theta_\pi}}[\tilde{\ell}_\pi^\tau(\theta_\pi,w,Q) + \lambda_1\ell_{\text{dep}}(w) + \lambda_2\ell_{\text{KL}}^\tau(\tilde{\tilde{\pi}}_{\theta_\pi},\tilde{\tilde{\pi}}_{\theta_{\pi_{\text{old}}}})] \\ \min_{\theta_v,w,Q} \mathbb{E}_{\tau\sim\tilde{\pi}_{\theta_\pi}}[\tilde{\ell}_v^\tau(\theta_v,w,Q)] \end{cases}. \tag{4}$$

Loss $\tilde{\ell}_\cdot(\theta,w,Q)$ originates from the original loss $\ell_\cdot(\theta)$ (Eqs. (1)-(2)) by transforming the deep net to include cross-connections, hence its dependence on $w,Q$. The tilde ('~') denotes this dependence, also for probability distribution $\tilde{f}$ and policy distribution $\tilde{\pi}$. Parameters from the current and a previous iteration are referred to via $\theta$ and $\theta_{\text{old}}$ respectively.

For both supervised and reinforcement learning, $\lambda_1$ and $\lambda_2$ control the strength which is used to decrease the influence of the teacher. A low $\lambda_1$ allows the student to rely on teachers. Close to the end of training, the student should be independent. Therefore, we set $\lambda_1$ to a small value at the beginning, and gradually increase its value as training progresses.

Note that we don't make any assumptions about teachers and student's objective. If a teacher's and student's objective differ, negative transfer may occur initially. However, the proposed method quickly decreases the weight for teacher layers to reduce this effect. Despite differences, students could potentially still benefit from the low level representation of the teachers. We do observe this low level knowledge transfer in our experiments.

In the following we first describe how to modify the deep nets, before we detail the loss functions $\ell_{\text{dep}}$ and $\ell_{\text{KL}}$, which are used to successively decrease the influence of the teachers.

## 3.2 DEEP NET TRANSFORMATION AND LOSS TERMS

**Deep Net Transformation:** *Knowledge flow* enhances the student by adding transformed and scaled intermediate representations from teacher models. To perform the transformation, intermediate representations from teachers are first multiplied by transformation matrices $Q$. Then the transformed representations from teachers and representations from the student are linearly combined. The weights for this linear combination are determined by a weight $p_w$ which is normalized to sum to one for each student layer.

Let index $m = 0$ denote the student model and let $\theta^{(0)}$ refer to its parameters. Further, let $\theta^{(m)}$, $m \in \{1, \ldots, M\}$ denote teacher models. We use $l_m^i$ to refer to deep net layer $i$ of teacher $m$, with $i \in \{1, \ldots, L_m\}$ and $L_m$ the number of layers in teacher $m$. We define layer $j$ of the student model to be $l_0^j$, where $j \in \{1, \ldots, L_0\}$ and $L_0$ the number of deep net layers in the student model. The output of layer $l_m^k$ right before and after an activation unit is denoted $z(l_m^k)$ and $h(l_m^k)$ respectively.

To align a teacher's layer $l_m^i$ with a student's layer $l_0^j$, we introduce a learnable transformation matrix $Q^j(l_m^i) \in \mathbb{R}^{\dim(l_0^j) \times \dim(l_m^i)}$, where $\dim(\cdot)$ gives the number of elements in the corresponding layer. The matrix multiplication $Q^j(l_m^i)z(l_m^i)$ aligns the representation from layer $i$ of teacher $m$ with the representation of layer $j$ of the student.

For each layer $j$ in the student model, we define a candidate set $\mathbb{L}^j$, which contains $l_0^j$ and all the teachers' layers to be considered. For example, in Fig. 1 (a), layer one of the student model is combined with layer one of teacher one and layer two of teacher two. Therefore, the candidate set of layer one of the student model is given by $\mathbb{L}^1 = \{l_0^1, l_1^1, l_2^2\}$.

To decide which teachers' or the student's representation to trust at every layer of the student net, we introduce a normalized weight $p_w^j(l)$ for all $j \in \{1, \ldots, L_0\}$, where $l \in \mathbb{L}^j$, summing to one for each layer $j$ in the student deep net, *i.e.*,

$$\sum_{l \in \mathbb{L}^j} p_w^j(l) = 1, \ \ \forall j \in \{1, \ldots, L_0\}.$$

To obtain the combined intermediate representation of layer $j$ for the student model, we use

$$h(l_0^j) = \sigma \left( \sum_{l \in \mathbb{L}^j \setminus l_0^j} p_w^j(l) Q^j(l) z(l) + p_w^j(l_0^j) z(l_0^j) \right),$$

where $p_w^j(l_m^i)$ determines how much the student layer $j$ relies on transformed representations of layer $i$ from the $m$-th teacher. Intuitively, if the transformed representation of the $m$-th teacher layer $i$ is helpful, $p_w^j(l_m^i)$ will be close to one. We visualize the deep net transformation in Fig. 1 (b).

Note that the intermediate representations of teachers are not changed in our framework. To obtain the output of layer $l_m^k$ we apply the original activation unit to the original representation $z(l_m^i)$, *i.e.*, $h(l_m^i) = \sigma(z(l_m^i)), \ \forall m \in \{1, \ldots, M\}, j \in \{1, \ldots L_m\}$.

The maximal number of introduced matrices $Q$ in our framework is $\sum_{i=1}^M L_i L_0$. In practice, we don't link a student's layer to every layer of a teacher network. Intuitively, a teachers' bottom layer

Table 1: Comparison with PathNet (Fernando et al., 2017) and progressive neural network (PNN) (Rusu et al., 2016b). Since PathNet and PNN don't report exact scores we obtain their numbers from their plots and indicate that with a $\sim$ symbol. The results of the state-of-the-art methods: A3C (Mnih et al., 2016), PPO (Schulman et al., 2017), and ACKTR (Wu et al., 2017) on Atari games are also listed for reference.

| | w/ Seaquest teacher | | w/ Riverraid teacher | | w/ Sea. and River. teachers | | No teachers | | |
|---|---|---|---|---|---|---|---|---|---|
| | Ours | PathNet | Ours | PathNet | Ours | PNN | A3C | PPO | ACKTR |
| Alien | 1254 | $\sim$**1700** | 1259 | $\sim$**1800** | 1911 | $\sim$**2000** | 182 | 1850 | 3197 |
| Asterix | **3982** | $\sim$2000 | **3823** | $\sim$2000 | 6012 | $\sim$**9000** | 6723 | 4533 | 31583 |
| Boxing | **96** | $\sim$70 | **96** | $\sim$80 | **99** | $\sim$**99** | 34 | 95 | 1 |
| Gopher | **4152** | $\sim$3900 | **3820** | $\sim$2100 | 5233 | $\sim$4500 | 8443 | 2933 | 47730 |
| Hero | **21250** | $\sim$12500 | **29343** | $\sim$12500 | **30928** | $\sim$30000 | 28766 | n/a | n/a |
| James. | **857** | $\sim$600 | **832** | $\sim$600 | **1245** | $\sim$850 | 352 | 561 | 512 |
| Krull | **8193** | $\sim$7800 | 6890 | $\sim$**7500** | **10000** | $\sim$9954 | 8067 | 7942 | 9689 |

features are very likely irrelevant to a student's top layer features. Indeed, we observed that linking a teachers' bottom layer to a student's top layer generally doesn't yield improvements. Therefore, in practice, we recommend to link one teacher layer to one or two student layers, in which case we introduce on the order of $ML_0$ matrices $Q$. Also note that while additional trainable parameters $Q$ and $w$ are introduced in our framework, $Q$ and $w$ are not part of the resulting student network since we ensure $p_w^j(l) \equiv 0 \; \forall l \in \mathbb{L}^j \backslash l_0^j$ at the end of training as discussed next. Hence, the additional parameters function as auxiliary knobs that help the student learn faster. In the final stage of training, the student will be independent (see Fig. 1 (c)) and does no longer rely on $Q$, $w$, or any transformed representations from teachers.

**Decreasing Teachers' Influence:** We successively decrease the influence of the teachers during training by gradually encouraging the normalized weight $p_w^j(l_0^j)$ to increase to a value of $1 \; \forall j \in \{1, 2, \ldots, L_0\}$. To capture how much the student relies on teachers, we introduce the *dependence cost* as the negative log probability:

$$\ell_{\text{dep}}(w) = -\frac{1}{L_0} \sum_{j \in \{1,2,\ldots,L_0\}} \log p_w^j(l_0^j). \tag{5}$$

By minimizing $\ell_{\text{dep}}(w)$, we encourage weights for the layers of the student to increase. Hence we encourage the student to become more and more independent. During the final stage of training, $p_w^j(l_0^j)$ approaches one for all $j \in \{1, \ldots, L_0\}$, making the student independent of the transformed representation obtained from teachers.

Empirically, we found that a fast decrease of the influence of the teacher can degrade the performance. This is intuitive as it requires some time to find good transformations $Q$. Moreover, decreasing the influence of a teacher too fast may change the output distribution over labels or actions of the student model too much, and thus lead to performance loss. To prevent changing a student's output distribution too fast, we found a Kullback-Leibler (KL) regularizer to yield good results. More specifically, in the case of supervised learning we use

$$\ell_{\text{KL}}(\tilde{\tilde{f}}_\theta, \tilde{\tilde{f}}_{\theta_{\text{old}}}) = D_{\text{KL}}[\tilde{\tilde{f}}_\theta(\cdot|x)||\tilde{\tilde{f}}_{\theta_{\text{old}}}(\cdot|x)]. \tag{6}$$

Hereby, $\theta$ is the set of current parameters, and $\theta_{\text{old}}$ are the previous ones. In the reinforcement learning case we use $D_{\text{KL}}[\tilde{\tilde{\pi}}_\theta(\cdot|x_t)||\tilde{\tilde{\pi}}_{\theta_{\text{old}}}(\cdot|x_t)]$.

## 4 EXPERIMENTAL RESULTS

In the following we evaluate *knowledge flow* on reinforcement and supervised learning tasks. Results are reported by using only the student model to avoid even the smallest influence from any teacher nets.

### 4.1 REINFORCEMENT LEARNING

We evaluate *knowledge flow* on reinforcement learning using Atari games that were used by Rusu et al. (2016b); Fernando et al. (2017). Following existing work, the input to our agent are raw images from the environment. The agent learns to predict actions only based on the rewards and the input

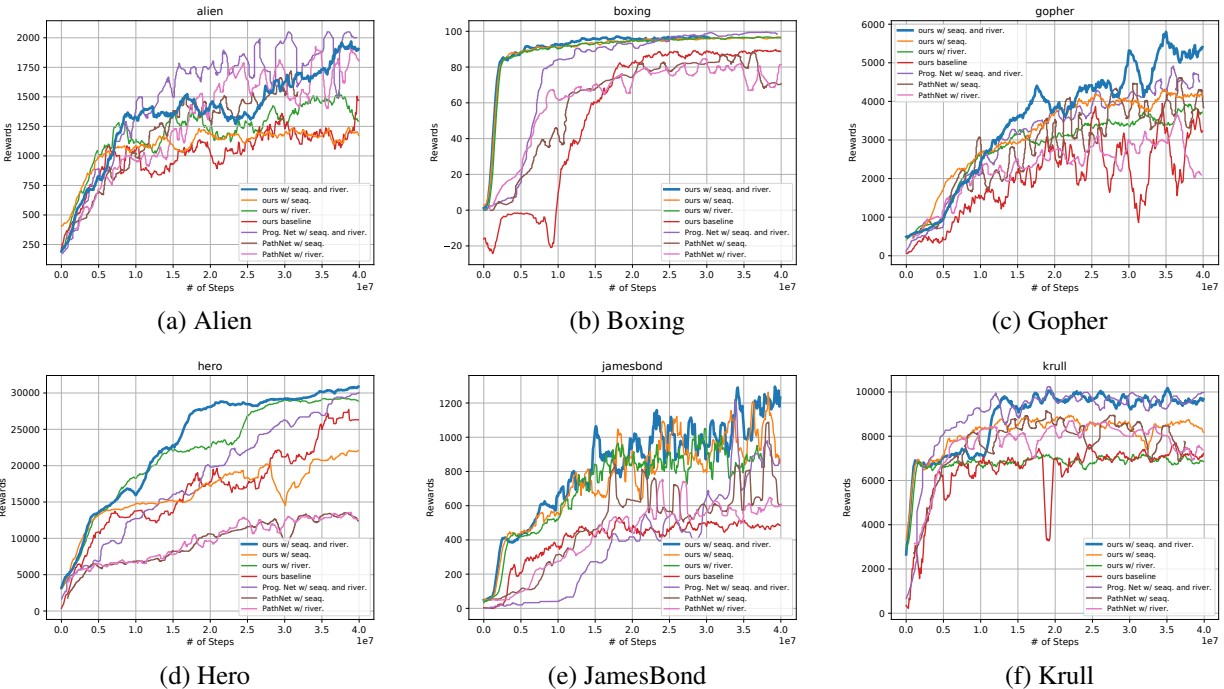

Figure 2: Comparison with progressive neural network and PathNet.

images from the environment. The agent chooses an action every four frames, and the last action is repeated on the skipped four frames. For all teacher models and the student model, we use the fully forward architecture of A3C (Mnih et al., 2016). The model has three hidden layers. The first layer is a convolutional layer with 16 filters of size 8x8 and stride 4. The second layer is a convolutional layer with 32 filters of size 4x4 and stride 2. The third layer is a fully connected layer with 256 hidden units. Following the third hidden layer are two sets of output. One is a softmax output that provides a probability distribution over all valid actions. The other one is a scalar output that provides the estimated value function. We use the same hyper-parameter settings as Mnih et al. (2016) except for the learning rate. Mnih et al. (2016) use RMSProp with shared statistics while we use Adam with shared statistics, which we found to give better results when training the baselines. The learning rate is set to $10^{-4}$ and gradually decreased to zero for all experiments. To select $\lambda_1$ and $\lambda_2$ in our framework, we follow progressive neural net (Rusu et al., 2016b): randomly sample $\lambda_1 \in \{0.05, 0.1, 0.5\}$ and $\lambda_2 \in \{0.001, 0.01, 0.05\}$. Note that $\lambda_1$ is set to zero at the beginning of training, and linearly increased to the sampled value at the end of training. Following Rusu et al. (2016b), we repeat each experiment 25 times with different random seeds and randomly sampled $\lambda_1$ and $\lambda_2$. The results of the top three out of 25 runs are reported. As A3C, we run 16 agents on 16 CPU cores in parallel.

**Evaluation Metrics:** We follow the evaluation procedure of Mnih et al. (2015). The trained student models are evaluated by playing each game for 30 episodes. We also follow the 'no-op' procedure: at the beginning of each testing episode, the agents perform up to 30 'no-op' actions.

**Results:** We first compare our framework with PathNet (Fernando et al., 2017) and progressive neural net (PNN) (Rusu et al., 2016b), which are state-of-the-art transfer reinforcement learning frameworks, using their experimental settings. The comparison is summarized in Table 1. The state-of-the-art results (Mnih et al., 2016; Schulman et al., 2017; Wu et al., 2017) on Atari games are also included in Table 1 for reference. Compared to PathNet, a student model trained using our transfer framework with one teacher achieves higher scores in 11 out of 14 experiments. Compared with PNN, for a two-teacher framework, our trained student model has only 0.7M parameters and PNN has 16M parameters. Nonetheless we observe higher scores in five out of the seven experiments. The results demonstrate that *knowledge flow* effectively transfers knowledge from teachers to the student. Table 1 also indicates that, in our framework, when the number of teachers increases from one to two, the student's performance improves significantly across all experiments. The training curves for the experiments are shown in Fig. 2. The curve is the average of the top three out of 25 runs. We observe our approach to generally perform very well.

Table 2: Comparison with fine-tuning and baseline A3C on different environment/teacher settings. The subscript following each number indicate the teachers being used. *E.g.*, (alien, space I.) indicates that one teacher is an alien expert and the other is a space invaders expert.

| | Ours w/ expert | Ours w/ non-expert | Fine-tune | A3C baseline | A3C |
|---|---|---|---|---|---|
| Alien$_{(teachers)}$ | **1705**$_{(alien, space I.)}$ | 1923$_{(bank., space I.)}$ | 996$_{(bank.)}$ | 1303$_{(n/a)}$ | 182$_{(n/a)}$ |
| Breakout$_{(teachers)}$ | 400$_{(breakout, space I.)}$ | 306$_{(pong, space I.)}$ | 261$_{(space I.)}$ | 99$_{(n/a)}$ | **552**$_{(n/a)}$ |
| ChopperCommand$_{(teachers)}$ | **8120**$_{(chopper., space I.)}$ | 6013$_{(sea., space I.)}$ | 3789$_{(sea.)}$ | 4513$_{(n/a)}$ | 4669$_{(n/a)}$ |
| KungFuMaster$_{(teachers)}$ | 29458$_{(kungfu., sea)}$ | **35103**$_{(sea. hero)}$ | 26752$_{(hero)}$ | 29446$_{(n/a)}$ | 3046$_{(n/a)}$ |
| MsPacman$_{(teachers)}$ | 2411$_{(mspac., alien)}$ | **2450**$_{(alien, space I.)}$ | 1324$_{(alien)}$ | 1628$_{(n/a)}$ | 594$_{(n/a)}$ |
| Seaquest$_{(teachers)}$ | 1873$_{(sea., chopper.)}$ | **32103**$_{(chopper., space I.)}$ | 1590$_{(chopper.)}$ | 1670$_{(n/a)}$ | 2300$_{(n/a)}$ |

(a) Seaquest      (b) KungFuMaster      (c) Alien

Figure 3: Comparison with fine-tuning and baseline A3C on different combinations of environment/teacher settings.

To further evaluate *knowledge flow*, we experiment with different combinations of environment/teacher settings. These settings are not used by PathNet and progressive neural network. The results are summarized in Table 2, where "ours w/ expert" represents that one teacher is expert for the target game; "ours w/ non-expert" represents that both teachers are not experts for the target game; "Fine-tune" represents fine-tuning from a non-expert on a new target game; "A3C baseline" represents our implementation of the A3C baseline; "A3C" represents the scores reported originally (Mnih et al., 2016). Note that our A3C implementation achieves better scores than those reported by Mnih et al. (2016) for most of the games. As shown in Table 2, *knowledge flow* with expert teacher performs better than the baseline across all experiments, which we interpret as evidence that *knowledge flow* successfully transfers 'knowledge' from an expert teacher to the student. In addition, *knowledge flow* with non-expert teachers also outperforms fine-tuning on a non-expert teacher. The reasons are twofold: First, a student model in *knowledge flow* can learn from multiple teachers while the fine-tuning method can only start from one setting. Second, in *knowledge flow*, the student can avoid the negative impact from insufficiently pretrained teachers, while fine-tuning from an insufficiently pretrained model slows down the training process and may degrade the overall performance. The training curves for the experiments are shown in Fig. 3. More training curves are in the Appendix (Fig. 6). Note that in *knowledge flow*, the student can benefit from the intermediate representations of the teacher, even if input space, output space and objectives differ. For example, in Fig. 3 (a), the two teachers are Chopper Command and Space Invaders, which are quite different from the target game Seaquest. The student model still benefits from learning from the teachers and achieves scores ten times larger than learning without teacher and fine-tuning from a teacher.

## 4.2 SUPERVISED LEARNING

For supervised learning, we use a variety of image classification benchmarks, including CIFAR-10 (Krizhevsky, 2009), CIFAR-100 (Krizhevsky, 2009), STL-10 (Coates et al., 2011), and EM-NIST (Cohen et al., 2017). The parameters $\lambda_1$ for the dependent cost and $\lambda_2$ for the KL cost are determined using the validation set of each dataset.

**Evaluation Metrics:** To evaluate the trained student model we report top-1 error rate on the test set of each dataset. All plots and reported numbers are the average of three runs obtained using different random seeds.

Table 3: Test Error (%) on CIFAR-10/100. The parentheses following "Ours" indicates the teachers we use. *I.e.*, 'Ours (SVHN, C100)' indicates that we use an SVHN expert and a C100 expert as teachers.

| | Baseline Densenet | Fine-tune from C100 | Fine-tune from SVHN | Ours (C100, SVHN) | | Baseline Densenet | Fine-tune from C10 | Fine-tune from SVHN | Ours (C10, SVHN) |
|---|---|---|---|---|---|---|---|---|---|
| C10 | 4.44 | 4.27 | 4.58 | **3.88** | C100 | 21.64 | 20.83 | 21.02 | **20.78** |
| | | (a) | | | | | (b) | | |

**CIFAR-10/CIFAR-100:** CIFAR-10 and CIFAR-100 datasets consist of colored images of size $32 \times 32$. CIFAR-10 (C10) has 10 classes and CIFAR-100 (C100) has 100 classes. For both dataset, the training and test sets contain 50,000 and 10,000 images respectively. We perform all experiments on CIFAR-10 and CIFAR-100 with standard data augmentation (Huang et al., 2017).

We use Densenet (Huang et al., 2017) (depth 100, growth rate 24) as a baseline and follow their hyper-parameter settings to train our baseline, teacher and student models. For our approach, we first train teachers on CIFAR-10, CIFAR-100, and SVHN (Netzer et al., 2011). We then train the student model using a different combination of teachers. We compare our results to fine-tuning and the baseline model. As shown in Table 3 (a), for the CIFAR-10 target task, fine-tuning from the CIFAR-100 expert improves $4\%$ over the baseline. Fine-tuning from the SVHN expert performs worse than the baseline model. Intuitively, for the CIFAR-10 target task, the CIFAR-100 deep net is a good teacher while a deep net trained with SVHN isn't. Presented with both good and inadequate teachers, *knowledge flow* improves by $13\%$ over the baseline. This demonstrates that *knowledge flow* can not only leverage a good teacher's 'knowledge,' but it can also avoid misleading influence. As detailed in Table 3 (b), the results are similar on the CIFAR-100 dataset.

To further demonstrate the properties of *knowledge flow*, additional results are in the appendix.

## 5  RELATED WORK

As mentioned before, 'knowledge' transfer has been considered using a variety of techniques. We briefly discuss related work in contrast to our approach in the following and defer details to Sec. 8.

**PathNet** (Fernando et al., 2017) enables multiple agents to train the same deep net while reusing parameters and avoiding catastrophic forgetting. In contrast to this formulation we consider availability of multiple pre-trained teacher nets.

**Progressive Net** (Rusu et al., 2016b) leverages transfer and avoids catastrophic forgetting by introducing lateral connections to previously learned features. Our discussed method uses similar lateral connections. However, in contrast to Rusu et al. (2016b), our method ensures independence of the student upon training, addressing a limitation in (Rusu et al., 2016b) where only a fraction of the capacity of the student is eventually utilized.

**Distral** a neologism combining 'distill & transfer learning' (Teh et al., 2017) considers joint training of multiple tasks. Multiple tasks share a 'distilled' policy which encodes common behavior between different tasks. While each worker addresses its own task, a shared policy encourages consistency between the policies. Different from Distral, which is a multi-task learning framework, *knowledge flow* addresses a single task, while in multi-task learning, multiple tasks are addressed at the same time. Hence, common for multi-task learning and *knowledge flow* is a transfer of information. However, in multi-task learning, information extracted from different tasks are shared to boost performance, while, in *knowledge flow*, the information of multiple teachers is leveraged to help a student learn better a single, new, previously unseen task.

Other related work includes **actor-mimic** (Parisotto et al., 2016), **learning without forgetting** (Li & Hoiem, 2016), **growing a brain** (Wang et al., 2017), **policy distillation** (Rusu et al., 2016a), **domain adaptation** (Pan & Yang, 2010; Long et al., 2015; Tzeng et al., 2015), **knowledge distillation** (Hinton et al., 2015) or **lifelong learning** (Chen & Liu, 2016). A more detailed discussion on related work is provided in Sec. 8 of the supplementary material.

## 6  CONCLUSION

We developed a general *knowledge flow* approach that permits to train a deep net from any number of teachers. We showed results for reinforcement learning and supervised learning, demonstrating improvements compared to training from scratch and to fine-tuning. In the future we plan to learn when to use which teacher and how to actively swap teachers during training of a student.

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

Table 4: Test error (%) of distilled student net.

|  | MNIST | MNIST w/o digit '3' | C100 | Imagenet |
|---|---|---|---|---|
| Student alone | 1.46 | 11.06 | 31.87 | 30.24 |
| KD Hinton et al. (2015) | 0.74 | 2.06 | 30.28 | 30.04 |
| Ours | **0.73** | **1.05** | **30.07** | **29.05** |

Table 5: Our approach on the EMNIST Letters dataset.

| Model (Teacher) | Test error(%) |
|---|---|
| Cohen et al. (2017) | 14.85 |
| Fine-tune from EMNIST digits | 9.04 |
| Baseline | 9.20 |
| Ours (EMNIST letters) | **7.13** |
| Ours (EMNIST half letters) | 8.13 |
| Ours (EMNIST digit) | 8.11 |

# 7 APPENDIX

## 7.1 SUPERVISED LEARNING

**Comparison with Knowledge Distillation:** We follow knowledge Distillation (KD) (Hinton et al., 2015) to distill knowledge from a larger model (teacher) to a smaller model (student). The student models have 50% - 5% parameters of the teacher models. Following their setup, we conduct experiments on MNIST, MNIST with digit '3' missing in the training set, CIFAR-100, and ImageNet. For MNIST and MNIST with digit '3' missing, following KD, the teacher model is an MLP with two hidden layers of 1200 hidden units, and the student model is an MLP with two hidden layers of 800 hidden units. For CIFAR-100, we use the model from Chen (2017) as teacher model. The student model follows the structure of the teacher, but the number of output channels of each convolutional layer is halved. For ImageNet, the teacher model is a 50-layer ResNet (He et al., 2016), and the student model is a 18-layer ResNet. The test error of the distilled student model are summarize in Table 4. Our framework has consistently better performance than KD, because the student model in our framework benefits not only from the output layer behavior of the teacher but also from intermediate layer representations of the teacher.

**EMNIST:**

The 'EMNIST Letters' dataset consists of images of size $28 \times 28$ pixels showing handwritten letters. It has 26 balanced classes. Each class contains lower and upper case letters. The training and test sets contain 124,800 and 20,800 images respectively. The 'EMNIST Digits' dataset consists of images of size $28 \times 28$ pixels showing handwritten digits. It has 10 balanced classes. The training and test sets contain 240,000 and 40,000 images respectively.

In this case we use the MNIST model from Chen (2017) as a baseline, teacher and student model. We trained teachers on EMNIST Digits, EMNIST Letters, and EMNIST Letters with only 13 classes. Our target task is EMNIST Letters. The student model is trained with different teachers and the results are compared to fine-tuning, the baseline model, and the state-of-the-art results on EMNIST. The results are summarized in Table 5. Compared to the baseline and fine-tuning, student learning in our framework with expert teacher (EMNIST Letters), semi-expert teacher (Half EMNIST Letters), and non-expert teacher (EMNIST Digits) all have better performance. In Fig. 4 we illustrate the accuracy over epochs for training of different models.

**STL-10:**

The STL-10 dataset consist of colored images of size $96 \times 96$ pixels. It has 10 balanced classes. The training set contains 5,000 labeled images and 100,000 unlabeled images. The test set contains 8,000 images. In our experiment, we only use the 5,000 labeled images for training.

We use the STL-10 model from Chen (2017) as our baseline, teacher and student model. We trained teachers on CIFAR-10 and CIFAR-100. We compare our results to fine-tuning and the baseline in Table 6. Note that STL-10 is very similar to CIFAR-10 and CIFAR-100. Therefore, both CIFAR-10 and CIFAR-100 are very good teachers. As shown in Table 6, compared to the baseline, fine-tuning a

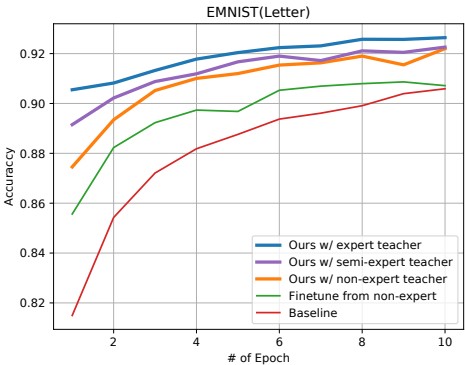

Figure 4: Comparison of top-1 accuracy of our approach, fine-tuning and baseline on the EMNIST Letters test dataset.

Table 6: Our approach on the STL-10 dataset (fully supervised).

|  | Test error (%) |
| --- | --- |
| Zhao et al. (2015) | 25.20 |
| Thoma (2017) | 21.34 |
| Baseline | 25.50 |
| Fine-tune from C10 | 14.32 |
| Fine-tune from C100 | 14.38 |
| Ours (C100) | 12.35 |
| Ours (C10, C100) | **11.09** |

model using weights pretrained on CIFAR-10 and CIFAR-100 reduce test errors by more than $10\%$. Compared with fine-tuning, student model training in our framework further reduces the test error by $3\%$. Note that we only train on the labeled data while other approaches use this data for testing of semi-supervised approaches. Hence our results are obtained using fewer data and may not be directly comparable. We still list their results in Table 6 for reference. In Fig. 5 we illustrate the accuracy over the epochs of training.

## 7.2 REINFORCEMENT LEARNING

We also compare to Distral (Teh et al., 2017), which is the state-of-the-art multi-task reinforcement learning framework. We used 'KL + ent 1 col', which has a central model ($m_0$), and a task model ($m_i$) for each task. We perform the experiments on Atari games. In the experiments, we have three tasks (task 1, task 2, task 3). The teachers of task 2 ($m_2$) and task 3 ($m_3$) are provided for our framework. Distral is trained for 120M steps (40M steps/task), and our model is trained for 40M steps. For fair comparison, we report results of Distral's task 1 model ($m_1$), which is better than its center model ($m_0$). The results are summarized in Table 7. Distral is suboptimal, because it aims to learn a multi-task agent. In addition, identical action and state space is assumed. When the target task is very different from the source tasks, Distral cannot decrease the teacher influence. In contrast, our framework can decrease a teacher's influence, and thus reduce negative transfer.

## 7.3 VISUALIZATION OF NORMALIZED WEIGHTS OF TEACHERS AND STUDENT

Following the reviewer's suggestion, we plot the averaged normalized weight ($p_w$) for teachers and the student in the C10 experiment, where C100 and SVHN experts are teachers. Intuitively, the C100 teacher should have a higher $p_w$ value than the SVHN teacher, because C100 is more relevant to C10. The plot verifies this intuition. As shown in Fig. 7, $p_w$ of the C100 teacher is higher than that of the SVHN teacher over the entire training. Note, both teachers' normalized weights approach zero at the end of training.

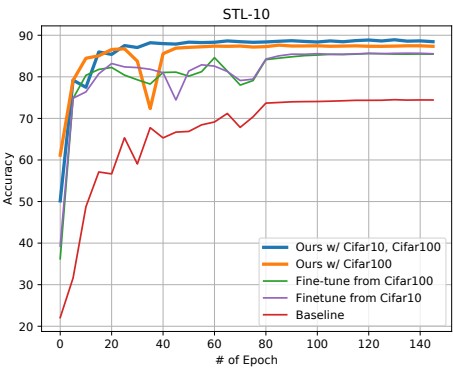

Figure 5: Comparison of top-1 accuracy of our approach, fine-tuning and baseline on the STL-10 test dataset.

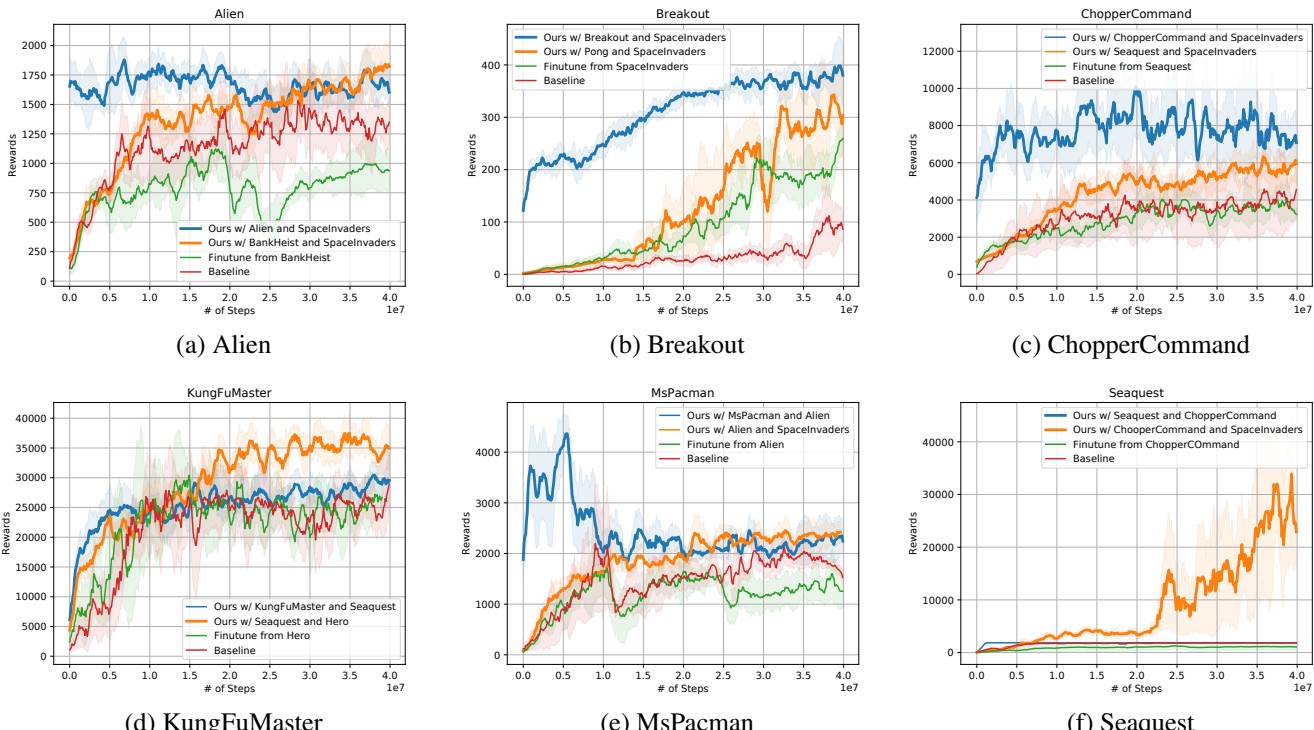

Figure 6: Comparison with fine-tuning and baseline A3C on different combinations of environment/teacher settings.

## 7.4 ABLATION STUDIES

### 7.4.1 UNTRAINED TEACHER MODELS

To verify that the student really benefits from the knowledge of teachers, we conduct an ablation study suggested by a reviewer. We use teacher models that haven't been trained at all. Intuitively, learning with untrained teachers should have worse performance than learning with knowledgeable teachers. Our experiments verify this intuition. In Fig. 8 (a), where the target task is hero, learning with untrained teachers ('w/ untrained teachers') achieves an average reward of 15934. Learning with knowledgeable teachers ('Ours with seaquest and riverraid teacher') achieves an average reward of 30928. More results are presented in Figs. 8 (b, c). The results show that *knowledge flow* achieves higher rewards than training with untrained teachers in different environments and teacher-student settings.

Table 7: Comparison with Distral on Task 1 score.

| Task1, Task2, Task3 | Distral Teh et al. (2017) | Ours |
|---|---|---|
| KungFuMaster, Hero, Seaquest | 27433 | **35103** |
| Hero, Seaquest, Riverraid | 15096 | **30928** |
| James, Seaquest, Riverraid | 550 | **1245** |

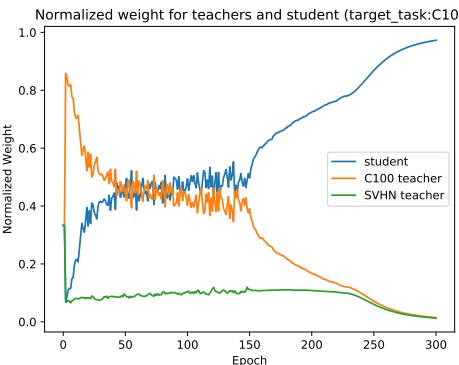

Figure 7: Normalized weights for the teachers and the student in C10 experiments.

### 7.4.2 TRAINING WITHOUT KL TERM

The KL term prevents the student's output distribution over actions or labels from drastic changes when the teachers' influence is decreasing. To investigate the importance of the KL term, we conduct an ablation study where the KL coefficient ($\lambda_2$) is set to zero. The result is summarized in Fig. 9. Considering Fig. 9 (a), where the target task is MsPacman and the teachers are Riverraid and Seaquest experts. Without the KL term, when a teacher's influence decreases, the rewards drop drastically. In contrast, with a KL term, we don't observe performance drops. At the end of training, learning with the KL term achieves an average reward of 2907 and learning without the KL term achieves an average reward of 1215. More results are presented in Fig. 9 (b, c), which shows that training with the KL term achieves higher reward than training without the KL term.

### 7.5 TEACHERS WITH DIFFERENT ARCHITECTURE THAN STUDENT

In additional experiments, following the suggestion of a reviewer, we use architectures for the teacher which differ from the student model. More specifically, we use the model of Mnih et al. (2015) as a teacher model. The teacher model consists of 3 convolutional layers, which have 32, 64, and 64 filters, followed by a hidden fully connected layer which has 512 ReLUs. We use the model of Mnih et al. (2016) as the student model. The student model consists of 2 convolutional layers, which have 16 and 32 filters respectively, followed by a hidden fully connected layer which has 256 ReLUs. Both models' fully connected layers are followed by two output layers for actions and values. In the experiments, we link each teacher's first convolutional layer to the student's first convolutional layer. Moreover, we link each teacher's third convolutional layer to the student's second convolutional layer, and each teacher's fully connected layer to the student's fully connected layer. In the experiment, the target task is KungFu Master, and the teachers are experts for Seaquest and Riverraid. The results are summarized in Fig. 10. We observed that learning with teachers, whose architecture differs from the student, to have similar performance as learning with teachers which have the same architecture. Consider as an example Fig. 10 (a), where the target task is KungFu Master, and the teachers are experts for Seaquest and Riverraid. At the end of training, learning with teachers of different architectures achieves an average reward of 37520, and learning with teachers of the same architecture achieves an average reward of 35012. More results are shown in Fig. 10 (b, c). The results show that *knowledge flow* can enable higher rewards, even if the teachers and the student architectures differ.

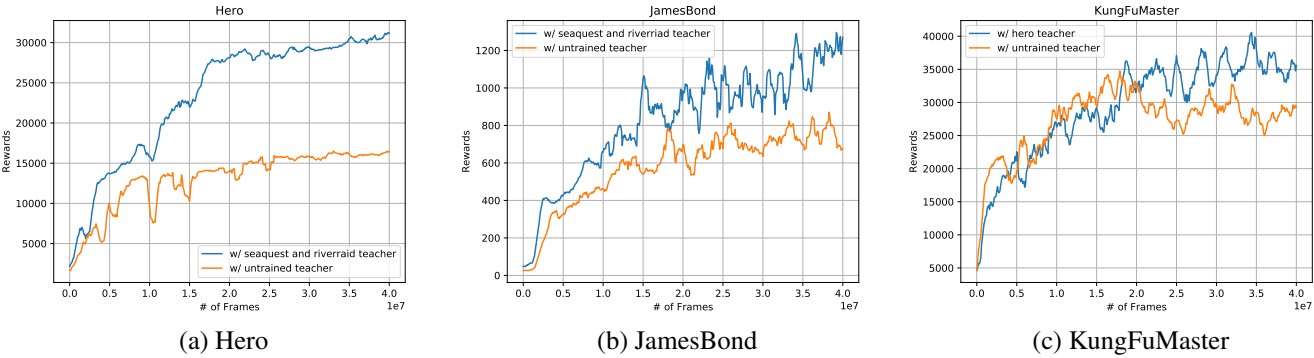

Figure 8: Ablation study: using untrained teachers.

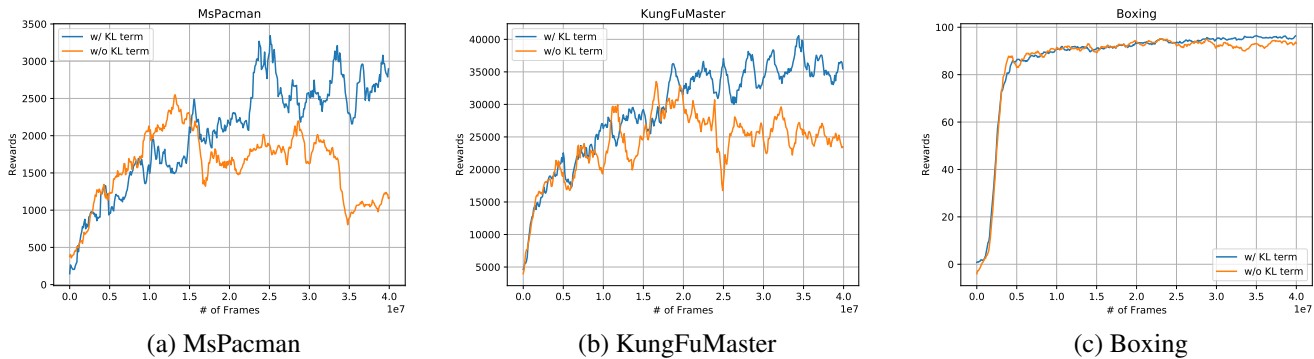

Figure 9: Ablation study regarding KL term. Seaquest and Riverraid experts are used as teachers for all experiments.

## 7.6 AVERAGE NETWORK AS $\theta_{old}$

For the parameters $\theta_{old}$ an average network can be used. To investigate how usage of an average network to obtain the parameters $\theta_{old}$ affects the performance, we conduct an experiment where $\theta_{old}$ is computed using the exponential running average of the model weight. More specifically, $\theta_{old}$ is updated as follows: $\theta_{old} \leftarrow \alpha \cdot \theta_{old} + (1 - \alpha) \cdot \theta$, where $\alpha = 0.9$. The results are summarized in Fig. 11. We observe that using an exponential average to compute $\theta_{old}$ results in very similar performance as using a single model. Consider Fig. 11 (a), where the target task is Boxing and the teacher is a Riverraid expert. At the end of training, using an average network to obtain $\theta_{old}$ achieves an average reward of 96.2 and using a single network to obtain $\theta_{old}$ achieves an average reward of 96.0. More results on using an average network are shown in Fig. 11 (b, c).

## 8 RELATED WORK

As mentioned before, variants of 'knowledge' transfer have been considered using a variety of techniques, for instance, fine-tuning, progressive neural nets (Rusu et al., 2016b), PathNet (Fernando et al., 2017), 'Growing a Brain' (Wang et al., 2017), actor-mimic (Parisotto et al., 2016), learning without forgetting (Li & Hoiem, 2016). Also related are techniques on transfer learning and lifelong learning. We discuss those methods and contrast them to our approach in the following.

**PathNet** (Fernando et al., 2017) enables multiple agents to train the same giant deep net while reusing parameters and avoiding catastrophic forgetting. To this end, agents embedded in the neural net discover which weights can be reused for new tasks and restrict application of gradients to those parameters. In contrast to this formulation we consider availability of multiple teacher nets, which are trained.

**Progressive Net** (Rusu et al., 2016b) leverages transfer and avoids catastrophic forgetting by introducing lateral connections to previously learned features. Our discussed method uses similar lateral

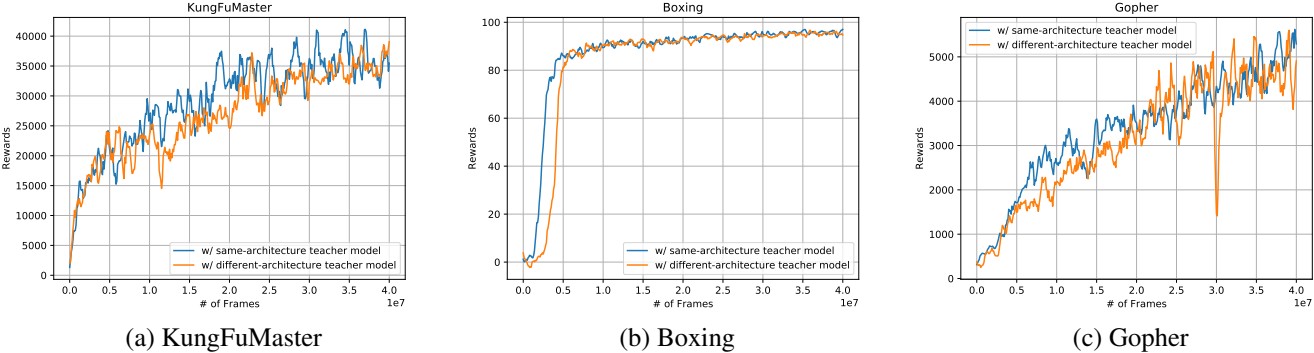

(a) KungFuMaster          (b) Boxing          (c) Gopher

Figure 10: Teachers' architecture differs from the student's architecture. Seaquest and Riverraid experts are used as teachers for all experiments.

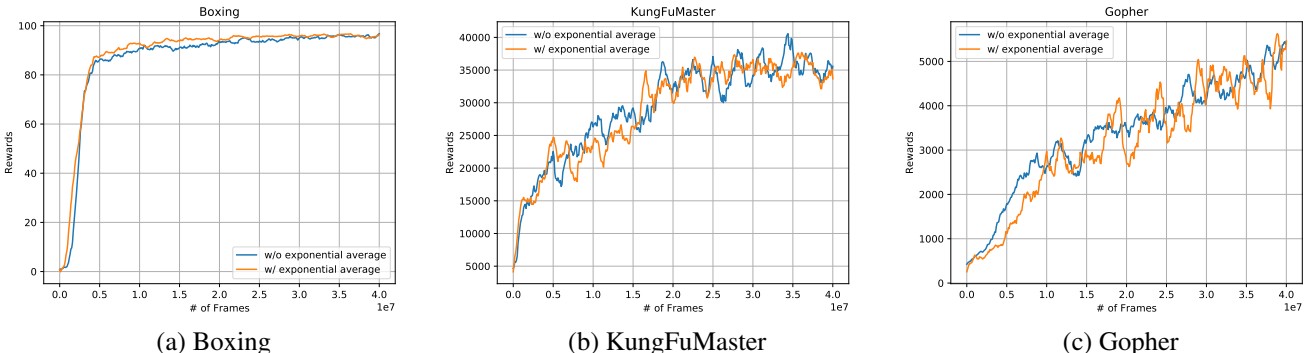

(a) Boxing          (b) KungFuMaster          (c) Gopher

Figure 11: Average network to compute $\theta_{old}$. Riverraid expert is used as teacher for all experiments.

connections. However, in contrast to Rusu et al. (2016b), we introduce scaling with normalized weights. This ensures independence of the student upon training, addressing a limitation in (Rusu et al., 2016b) where only a fraction of the capacity of the student is eventually utilized.

**Distral** a neologism combining 'distill & transfer learning' (Teh et al., 2017) considers joint training of multiple tasks. Multiple tasks share a 'distilled' policy which encodes common behavior between different tasks. While each worker addresses its own task, a shared policy encourages consistency between the policies. Different from Distral, which is a multi-task learning framework, *knowledge flow* addresses a single task, while in multi-task learning, multiple tasks are addressed at the same time. Hence, common for multi-task learning and *knowledge flow* is a transfer of information. However, in multi-task learning, information extracted from different tasks are shared to boost performance, while, in *knowledge flow*, the information of multiple teachers is leveraged to help a student learn better a single, new, previously unseen task.

**Knowledge distillation** (Hinton et al., 2015) distills information form a larger deep net into a smaller one. It assumes both nets are trained on the same dataset. In contrast, our technique allows knowledge transfer between different source and target domains.

**Actor-mimic** (Parisotto et al., 2016) enables an agent to learn how to address multiple tasks simultaneously and generalize the extracted knowledge to new domains. A single policy net learns how to act in a set of tasks following the guidance of several expert teachers. A combination of feature regression and cross entropy loss is used to encourage the student to produce similar actions and representations. Our proposed technique differs in that we take advantage of a teachers representation at the beginning of training,

**Learning without forgetting** (Li & Hoiem, 2016) permits to add a new task to a deep net without forgetting the original capabilities. Importantly, only data from the new task is used and the old capabilities are retained by first recording the old networks output on the new data. Similar techniques have been developed by Furlanello et al. (2016); Jung et al. (2016). In contrast, we transfer 'knowledge' from teacher networks more explicitly.

**Growing a Brain** (Wang et al., 2017) analyzes the parameters which change during fine-tuning and points out that more natural model adaptation is obtained when increasing the model capacity, by either extending width or depth. Appropriate normalization is essential to significantly outperform classical fine-tuning. Since this technique is based on fine-tuning, it differs from our student-teacher based approach.

Other related work includes **policy distillation** (Rusu et al., 2016a), **domain adaptation** (Pan & Yang, 2010; Long et al., 2015; Tzeng et al., 2015) or **lifelong learning** (Chen & Liu, 2016; Thrun, 1998; Mitchell et al., 2015; Ruvolo & Eaton, 2013).

