# OpenReview forum: "Knowledge Flow: Improve Upon Your Teachers"
_ICLR.cc/2019/Conference_

### Official Review · AnonReviewer3 · 2018-11-03
**interesting approach in combining multiple trained models for transfer**

**Rating:** 7
**Confidence:** 4

**Review:**

This paper presents a method for distilling multiple teacher networks into a student, by linearly combining feature representations from all networks at multiple intermediate layers, and gradually forcing the student to "take over" the learned combination.  Networks to be used as teachers are first pretrained on various initial tasks.  A student network is then trained on a target task (possibly different from any teacher task), by combining corresponding hidden layers from each teacher using learned linear remappings and weighted combinations.  Learning this combination allows the system to find appropriate teachers for the target task; eventually, a penalty on the combination weights forces all weight onto the student network, resulting in the distillation.

Applications to both reinforcement learning (atari game) and supervised image classification (cifar, svhn) are evaluated.  The reinforcement learning application is particularly fitting, since combining tasks together is less straightforward in this domain.

I wonder whether any experiments were performed where the layers correspondence between teacher models was less clear --- say, using teachers with different architectures.  Figure 1(a) (different teacher archs) as well as the text ("candidate set" on p.4) indicate this is possible, but experiment details describe combinations of same-architecture teachers only.

In addition, I would have liked to see some further exploration of the KL term and use of "theta_old".  This seems potentially important, and also has ties to self-ensembling through teachers with exponential weight averaging.  Could an average network also be used here?  And how important is this term in linking student to teachers as the weights change?

Overall I find this a very interesting approach.  Rather than training a large joint model on multiple tasks simultaneously as a transfer initialization, this approach uses models already fully trained for different tasks.  This results in a potentially advantageous trade-off:  One no longer needs to carefully calibrate the different tasks and common task components in a joint model, but at the cost of requiring inference through multiple teachers when training the student.

---

> ### Author Response · Authors · 2018-11-24
> **Response to AnonReviewer3:**
>
> Updated: Changed section numbers to fit latest revision.
> ---------------------------------------------------------------------------
> We thank the reviewer for time and feedback.
>
> Re 1: Use teachers with different architectures from the student.
> In additional experiments, following the suggestion of the reviewer, we use architectures for the teacher which differ from the student model. The results are summarized in Fig. 10 and discussed in Sec. 7.4. We observed that learning with teachers, whose architecture differs from the student, to have similar performance as learning with teachers which have the same architecture. Consider Fig.10 (a) as an example, where the target task is KungFu Master, and the teachers are experts for Seaquest and Riverraid. At the end of training, learning with teachers of different architectures achieves an average reward of 37520, and learning with teachers of the same architecture achieves an average reward of 35012. More results are shown in Figs. 10 (b, c). The results illustrate that Knowledge Flow can benefit from the knowledge of teachers, and thus achieve higher rewards, even if the teachers and the student architectures differ.
>
> Re 2: Importance of KL term.
> The KL term prevents the student’s output distribution over actions or labels to change too much when the teachers’ influence is decreasing. To investigate the importance of the KL term, we conduct an ablation study where the KL coefficient (\lambda2) is set to zero. The results are summarized in Fig. 9 and discussed in Sec. 7.3.2. Considering Fig. 9 (a) as an example, where the target task is MsPacman and the teachers are Riverraid and Seaquest experts. Without the KL term the rewards drop drastically when the teacher’s influence decreases. In contrast, we don’t observe this performance drop with a KL term. At the end of training, learning with a KL term achieves an average reward of 2907 and learning without the KL term achieves an average reward of 1215. More results are presented in Figs. 9 (b, c), which show that training with the KL term achieves higher reward than training without the KL term.
>
> Re 3: Use of an average network as \theta_{old}.
> An average network, i.e., exponential averaging can be used to obtain \theta_{old}. To investigate how usage of an average network for \theta{old} affects the performance, we conduct an experiment setting \theta_{old} to be the exponential running average of the model weight. More specifically, \theta_{old} is updated as follows: \theta_{old} \leftarrow \alpha * \theta_{old} + (1 - \alpha) * \theta, where \alpha = 0.9. The results are summarized in Fig. 11 and discussed in Sec. 7.5. We observed that using an exponential average to compute \theta_{old} results in very similar performance to using a single model. Consider Fig.11 (a) as an example, where the target task is Boxing and the teacher is a Riverraid expert. At the end of training, computing \theta_{old} via an exponential average achieves an average reward of 96.2 and using a single parameter to set \theta_old achieves an average reward of 96.0. More results on using an exponential average to compute \theta_{old} are shown in Figs. 11 (b, c).

---

### Official Review · AnonReviewer1 · 2018-11-03
**Intriguing idea, strong performance, but missing empirical results to validate intuition**

**Rating:** 8
**Confidence:** 5

**Review:**

This paper proposes to feed the representations of various external "teacher" neural networks of a particular example as inputs to various layers of a student network.
The idea is quite intriguing and performs very well empirically, and the paper is also well written.  While I view the performance experiments as extremely thorough, I believe the paper could possibly use some additional ablation-style experiments just to verify the method actually operates as one intuitively thinks it should.

Other Comments:

- Did you verify that in Table 3, the p_w values for the teachers trained on the more-relevant C10/C100 dataset are higher than the p_w value for the teacher trained on the SVHN data?  It would be interesting to see the plots of these p_w over the course of training (similar to Fig 1c) to verify this method actually operates as one intuitively believes it should.

- Integrating the teacher-network representations into various hidden layers of the student network might also be considered some form of neural architecture search (NAS)  (by including parts of the teacher network into the student architecture).
See for example the DARTS paper: https://arxiv.org/abs/1806.09055
which similarly employs mixtures of potential connections.
Under this NAS perspective, the dependence loss subsequently distills the optimal architecture network back into the student network architecture.

Have you verified that this method is not just doing NAS, by for example, providing a small student network with a few teacher networks that haven't been trained at all? (i.e. should not permit any knowledge flow)

- Have the authors considered training the teacher networks jointly with the student? This could be viewed as teachers learning how to improve their knowledge flow (although might require large amounts of memory depending on the size of the teacher networks).

- Suppose we have an L-layer student network and T M-layer teacher networks.
Does this imply we have to consider O(L*M*T) additional weight matrices Q?
Can you comment on the memory requirements?

- The teacher-student setup should be made more clear in Tables 1 and 2 captions (took me some time to comprehend).

- The second and third paragraphs are redundant given the Related Work section that appears later on. I would like to see these redundancies minimized and the freed up  space used to include more results from the Appendix in the main text.

---

> ### Author Response · Authors · 2018-11-24
> **Response to AnonReviewer1:**
>
> Updated: Changed section numbers to fit latest revision.
> ---------------------------------------------------------------------------
> We thank the reviewer for time and feedback.
>
> Re 1: plot p_w values for C10/C100 dataset.
> In the newly added Fig. 4 and the corresponding discussion (Sec. 4.2), we plot the weight (p_w) for teachers and the student in the C10/C100 experiment, where C100 and SVHN experts are teachers. As expected and intuitively, the C100 teacher should have higher p_w value than the SVHN based teacher, because C100 is more relevant to C10. The plot verifies this intuition, p_w of the C100 teacher is higher than that of the SVHN teacher during the entire training. Both teachers’ normalized weights approach zero at the end of training.
>
> Re 2: verify Knowledge Flow is not just NAS.
> As the reviewer pointed out, one key difference between NAS and Knowledge Flow is that a student in Knowledge Flow benefits from teachers’ knowledge. To verify that the student really benefits from the knowledge of teachers, we conduct the ablation study suggested by the reviewer. In the newly added experiment, discussed in Sec. 7.3.1 and summarized in Fig. 8, teachers are models that haven’t been trained at all. Intuitively, learning with untrained teachers should have worse performance than learning with knowledgeable teachers. Our experiments verify this intuition. Considering Fig. 8 (a), where the target task is hero, learning with untrained teachers achieves an average reward of 15934, while learning with knowledgeable teachers (experts of Seaquest and Riverraid) achieves an average reward of 30928. Consistently with all other experiments we average over five runs. More results are presented in Fig. 8 (b, c). The results show that Knowledge Flow achieves higher rewards than NAS in different environments and teacher-student settings.
>
> Re 3: training teacher networks jointly.
> We did try to train teachers jointly with students. However, as the reviewer mentioned, the memory usage is large and training is very slow. Up until now we didn’t observe any improvements.
>
> Re 4: memory requirement for matrices Q.
> The upper bound of the number of Q matrices in our framework is O(L*M*T). In practice, we don’t link a student’s layer to every layer of a teacher network. For example, we observed that linking a teachers’ bottom layer to a student’s top layer generally doesn’t yield improvements. Intuitively, a teachers’ bottom layer features are very likely irrelevant to a student’s top layer features. Therefore, in practice, we recommend to link one teacher layer to one or two student layers, in which case the space complexity is O(L*T).
>
> Re 5: Captions of Table 1 and Table 2.
> We updated the caption of Table 1 and Table 2.
>
> Re 6: Shorten paragraph 2 and paragraph 3.
> We felt shortening paragraph 2 and 3 would remove the motivation of this work. Shortening the related work section wouldn’t do justice to our peers. Therefore at this point we prefer to maintain the current writing unless the majority of the reviewers and the AC feel strongly about shortening.

---

> > ### Comment · AnonReviewer1 · 2018-11-26
> > **Nice revision!**
> >
> > The new additions to the paper are very welcome, and definitely make the paper stronger in my opinion.
> >
> > Re 4: I recommend the authors include this statement somewhere in the paper/appendix.
> >
> > Re 6: If the authors feel paragraphs 2 & 3 are critical to motivate this work, then I still think you can instead shorten the parts of the related work section to make the information there less redundant.  In my opinion, the main text would be better if you made space for Fig 7 from the appendix and the relevant text description, by reducing the redundancy in descriptions of the alternative methods and their shortcomings.

---

> > > ### Author Response · Authors · 2018-11-27
> > > **Response to AnonReviewer1:**
> > >
> > > Thanks a lot for additional time and feedback.
> > >
> > > Re 4: We added the comment regarding space complexity to Sec. 3.2 of the main paper.
> > >
> > > Re 6: We moved the detailed treatment of related work to the appendix and provide a shortened version in the main paper. We also moved Fig. 7 and the corresponding text to Sec. 4.2 of the main paper.

---

### Official Review · AnonReviewer2 · 2018-11-04
**Multiple task learning for NNs**

**Rating:** 6
**Confidence:** 3

**Review:**

This paper proposes a new set of heuristics for learning a NN for generalising a set of NNs trained for more specific tasks. This particular recipe might be reasonable, but the semi-formal flavour is distracting. The issue of model selection (clearly the main issue here) is not addressed. A quite severe issue with this report is that the authors don't report relevant learning results from before (+-) 2009, and empirical comparisons are only given w.r.t. other recent heuristics. This makes it for me not possible to advice publication as is.

---

> ### Author Response · Authors · 2018-11-24
> **Response to AnonReviewer2:**
>
> We thank the reviewer for time and feedback. We think the questions aren’t precise enough for us to act upon:
> 1. We’d appreciate if the reviewer can point out the parts that are according to the reviewer’s opinion `semi-formal’? We are more than happy to revise the text but are currently left guessing, particularly since another reviewer points out that the paper is `well written.’
> 2. We compare to recent baselines, in particular state-of-the-art methods like PNN and PathNet. If the reviewer would specify which papers from before 2009 we should compare to, we are very happy to include a statement, assuming that PNN and/or PathNet or their predecessors haven’t compared to those already.
> 3. To the best of our knowledge, the two baselines (PNN and PathNet) we compare with are the state-of-the-art RL transfer frameworks.

---

> > ### Comment · AnonReviewer2 · 2018-11-24
> > **Multiple-task learning for DNN**
> >
> > 2 and 3 are the same.
> >
> > Multiple-task learning approaches are rife in this area (see e.g. https://en.wikipedia.org/wiki/Multi-task_learning, and citations therein). This huge body of work establishes that using a proper regularisation scheme is central. The intuition in the present paper seems to align with those ideas. But since those are so standard by now, the authors can be  expected to make the connection explicit. Note that the idea of 'lifelong learning' as cited does acknowledge this connection.
> >
> > Multi-task learning for DNN is a standard theme (especially in this conference), and it is not clarified how this work relates/improves over this body of work. One way to address this issue is to report empirical results on a standard benchmark (as MNIST).
> >
> > The introductory text (ch, 2) is not quite correct (especially the RL needs care), but can be patched up by citing relevant introductory texts (what is random etc..) and adhering to their notation.

---

> > > ### Author Response · Authors · 2018-11-26
> > > **Response to AnonReviewer2:**
> > >
> > > Thanks a lot for additional time, feedback and clarifications.
> > >
> > > Re 1: Multi-task learning.
> > > Note that the challenge we address differs from multi-task learning. In multi-task learning, multiple tasks are addressed at the same time. In contrast, `Knowledge Flow’ focuses on a single task. Hence, common for multi-task learning and `Knowledge Flow’ is a transfer of information. However, in multi-task learning, information extracted from different tasks are shared to boost performance, while, in `Knowledge Flow,’ the information of multiple teachers is leveraged to help a student learn better a single, new, previously unseen task. We updated Section 5 to clarify the connection and differences.
> > >
> > >
> > > Re 2: Notation of Section 2.
> > > We follow the notation of Mnih et al. (2016), i.e., the expectation is taken with respect to a trajectory \tau = ({x_t, a_t, r_t}, {x_{t+1}, a_{t+1}, r_{t+1}}, ...) generated by following the policy \pi. We clarified this and updated Section 2 and 3.

---

### Public Comment · (anonymous) · 2019-07-05
**The additional memory cost of large deep network.**

Hi, authors,

Thanks for this interesting work. I have a question about the size of matrix Q. One Q's dimension is (the size of a teacher's feature map)x(the size of student's feature map). For ImageNet, one intermediate feature map might be 512x14x14= 100352. Is Q becomes a 100352 x 100352 matrix? If so, it would be too large to run.

In addition, would you mind to release codes?

Best Regards,

---

### Meta-Review · Area_Chair1 · 2018-12-15

**Confidence:** 5
**Recommendation:** Accept (Poster)

**Metareview:**

The authors have taken inspiration from recent publications that demonstrate transfer learning over sequential RL tasks and have proposed a method that trains individual learners from experts using layerwise connections, gradually forcing the features to distill into the student with a hard-coded annealing of coeffiecients. The authors have done thorough experiments and the value of the approach seems clear, especially compared against progressive nets and pathnets. The paper is well-written and interesting, and the approach is novel. The reviewers have discussed the paper in detail and agree, with the AC, that it should be accepted.